# Spatiotemporal Characteristics of Land Cover Change in the Yellow River Basin over the Past Millennium

**Yafei Wang** [1,2], **Fan Yang** [3,*] **and Fanneng He** [1]

1    Key Laboratory of Land Surface Pattern and Simulation, Institute of Geographic Sciences and Natural Resources Research, Chinese Academy of Sciences, Beijing 100101, China; wangyafei972x@igsnrr.ac.cn (Y.W.); hefn@igsnrr.ac.cn (F.H.)
2    University of Chinese Academy of Sciences, Beijing 100049, China
3    Key Research Institute of Yellow River Civilization and Sustainable Development, Collaborative Innovation Center on Yellow River Civilization Jointly Built by Henan Province, Ministry of Education, Henan University, Kaifeng 475001, China
*    Correspondence: yangfan@henu.edu.cn

**Abstract:** Investigating the ecological and environmental impacts stemming from historical land use and land cover change (LUCC) holds paramount importance in systematically comprehending the fundamental human-land relationship, a pivotal focus within geographical research. The Yellow River Basin (YRB), often referred to as the cradle of Chinese civilization, ranks as the fifth-largest river basin globally. Early inhabitants made significant alterations to the landscape, resulting in substantial damage to natural vegetation, giving rise to prominent regional ecological challenges. By now, the examination of historical LUCC in the YRB over the past millennium remains in the qualitative research stage, primarily due to the limited availability of high-confidence gridded historical LUCC data. This study aims to advance the current historical LUCC research in the YRB from primarily qualitative analysis to an exploration incorporating timing, positioning, and quantification. Based on reconstructed historical cropland, forest, and grassland grid data of 10 km × 10 km from 1000 AD to 2000 AD, the degree of cropland development and the depletion of forests and grasslands were calculated, respectively. Then, the kernel density method was employed for spatiotemporal analysis and interpretation of dynamic changes in land cover. Subsequently, a cartographic visualization depicting the migration trajectories of the land cover gravity centers was generated, allowing for an assessment of the distance and direction of the centroids' movement of cropland, forest, and grassland. The results indicate that the cropland coverage in the YRB escalated from the initial 11.65% to 29.97%, while the forest and grassland coverage dropped from 63.36% to 44.49%. The distribution of cultivated land continually expanded outward from the southeast of the Loess Plateau and the southwest of the North China Plain. All three types of land cover experienced a westward shift in their gravity centers between 1000 and 2000 AD. Besides the population growth and technological advancements, the regime shifts induced by wars, along with land use policies in distinct periods, always served as the predominant factors influencing the conversion between different land covers. This research will present a paradigmatic regional case study contributing to the investigation of historical changes in land use and land cover. Additionally, it will offer historical perspectives beneficial for the advancement of China's objectives in "Ecological Conservation and High-Quality Development of the Yellow River Basin".

**Keywords:** land use and land cover change (LUCC); spatiotemporal analysis; past millennium; the Yellow River Basin

## 1. Introduction

Anthropogenic activities exert a predominant influence on natural ecosystems by modulating environmental dynamics primarily through alterations in land use patterns. Archaeological evidence indicates that substantial alterations to the Earth's surface stem

from human activities commenced approximately 3000 years ago [1,2]. This transformation has led to more than 37% of the world's ice-free land being currently domesticated [3–5]. Over historical epochs, human activities have consistently influenced the environment by altering land cover and disrupting landscape succession, which has aroused attention from researchers toward historical LUCC [6–9]. Extensive quantitative reconstructions of long-term LUCC on both global [5,10–12] and regional scales [13–18] have been achieved with the support of various scientific projects, such as the Past Global Changes, Land Cover 6k, and Global Land Project. These studies, exploring the historical dynamics of human-land relationships, have the potential to enhance the cross-disciplinary integration between LUCC and allied fields, including historical climate change [19,20] and the carbon cycle [21–23].

The Yellow River Basin, acclaimed as the cradle of Chinese civilization, reached its zenith during China's feudal society, emerging as the most densely populated, politically, economically, and culturally advanced region [24]. It stands among the earliest regions in China to embrace agricultural activities, and the engagement between inhabitants and the river basin is one of the longest and most intensive globally. However, persistent human interference (agriculture and deforestation) has significantly altered the natural vegetation cover, causing irreversible impacts on the local environment [25,26]. These accumulated impacts, spanning an extended period, have consistently influenced the socio-economic development of this area, particularly over the past millennium since the Song Dynasty [27]. This period has witnessed severe repercussions linked to environmental changes and ecological security, notably including issues such as droughts, floods, soil erosion in the Loess Plateau, and downstream water hazards [28–31]. In 2019, the ecological conservation and high-quality development of the Yellow River Basin evolved into a key national strategy in China [32]. In 2022, the Ministry of Ecology and Environment issued the "Ecological Environment Protection Plan for the Yellow River Basin," aligning with the goals of the Beautiful China Initiative. Consequently, studies pertaining to land use and ecological environmental protection in the YRB have regained prominence within the academic community, garnering widespread attention.

Significant advancements have been achieved in the comprehensive qualitative analysis of historical LUCC within the YRB. This progress is grounded in the meticulous examination of extensive historical inventory data. In history, population [33] growth and agricultural practices led to extensive destruction of forest and grassland vegetation in the middle reaches of the YRB [26]. The natural vegetation underwent a transition from abundance to scarcity and, in some cases, disappearance [34]. The depletion of natural vegetation has resulted in escalated soil erosion, causing a swift augmentation of the sediment discharge [35]. This, in turn, has contributed to the continuous aggradation of the riverbed downstream and a heightened frequency of flooding, along with more frequent incidents of river breaches and diversions in the lower reaches [36]. Zheng et al. [26] conducted a thorough assessment of the characteristics of climate change and the overarching trend of land cover generated by agricultural land use in the YRB over the past two thousand years. Wu et al. [37] opted for climate, socioeconomic, and policy indicators to analyze the local and spillover effects resulting from social-ecological interactions in various regimes within the Loess Plateau, situated in the central region of the YRB. The interactions among population, forest coverage, and cropland area encapsulate the intricate relationship between societal development and the dual imperatives of food supply and environmental sustainability. These investigations present valuable insights and compelling illustrations for examining the historical dynamics of human-land relationships and the impact of human activities on the ecological environment. However, there is still a deficiency in providing spatially explicit representations of dynamic LUCC in the YRB during historical periods, particularly in cartographic form.

Advancements in methodology and model development have facilitated the progress in the gridded reconstruction of land cover. At the same time, the existing research indicates significant uncertainty in space and time in the reconstruction results of global LUCC

datasets at the regional scale [10,38–42]. Furthermore, the nationwide data covering the YRB is frequently subjected to extensive analysis for a singular land-use type on a national scale, lacking more detailed studies specifically addressing the YRB [16,34,43,44]. In the past decade, significant progress has been achieved on spatially explicit historical land cover reconstruction for the YRB. These investigations commonly integrate historical documents and contemporary remote sensing data to reconstruct the gridded data of land cover, concentrating on scrutinizing alterations in reclaimed land rates. Hou et al. (2012) [45] utilized GIS tools and Site Domain Analysis (SDA) methods to reconstruct the mid-Holocene cultivated land in the midstream and downstream areas. Luo et al. (2014) [46] produced gridded maps of cropland distribution in the Yellow River-Huangshui River Valley (YHV), located at the upper reaches of the Yellow River, in 1726 AD. Wu et al. (2016) [47] integrated a contemporary vegetation map with the reconstructed farmland change from the Qing Dynasty to estimate the changed area and spatial distribution of forest land, grassland, and shrubland in YHV. Tian et al. (2012) [48] applied Exploratory Spatial Data Analysis (ESDA) to examine the temporal changes in the cropland area across counties in the Loess Plateau over the last 300 years. Ye et al. (2015) [49] presented data sourced from 244 local gazetteers, government statistical records, and remote-sensing land cover data to reconstruct the county-level reclaimed land for Shandong Province. Wei et al. (2019, 2021) [50,51] combined available datasets and historical statistics to reconstruct the spatially explicit cropland for the North China Plain Area (10 km resolution, 17th century to the 1980s) and the Guanzhong Area in the YRB (1 km resolution, 1650 AD to 2016 AD). Yang et al. (2024) [35] reconstructed forest coverage data, showing deforestation expansion from plains to surrounding hills and mountainous areas in the mid-to-lower reaches.

Summarizing the findings of the studies mentioned above, the research domain primarily encompasses specific regions within the contemporary YRB, encompassing the Loess Plateau [48,52], North China Plain [50,51], the middle and lower reaches [35,45], Hehuang Valley [46,47,53,54], and provinces [49]. The temporal scope spans from the mid-Holocene to recent centuries or a particular historical year. Previous work has provided a dependable database for examining the long-term dynamics of coupled natural-human systems and simulating climate and environmental changes within the YRB. However, in historical contexts, the spatial extent of the lower reaches of YRB exceeds the contemporary boundaries due to factors such as flooding, river breaches, and diversions within the lower reaches. The existing studies are constrained in their focus on specific segments of the YRB, overlooking the intricate interactions and internal relationships among its upper, middle, and lower reaches. Furthermore, these findings are incapable of providing an objective reflection of the dynamic changes among diverse land use categories across the YRB. In summary, there is still an absence of depiction of the long-term sequential transformations and an analysis of the driving forces propelling these dynamic changes.

Here, we have redefined the historical extent of the lower reaches of YRB by examining its oscillations and alterations over the past millennium. Then, we obtained the grid data for cropland, forest land, and grassland by utilizing the Extract by Mask tool in ArcGIS, with the YRB boundaries serving as the basis. We employed methodologies to systematically quantify changes in land cover and elucidate the mechanisms governing the spatiotemporal evolution in the YRB over the past millennium. The objective of this study is to fill the gap in the exploration of historical LUCC within the YRB over the past millennium that integrates temporal, spatial, and quantitative dimensions. Subsequently, the investigation of the driving forces influencing the spatiotemporal changes in land cover during different historical periods will be conducted. Because the past is inextricably bound to the present and future, this study anticipates offering historical insights to comprehend the current environmental issues and implications for future high-quality and sustainable development planning in the YRB.

## 2. Study Area and Data Sources

### 2.1. Study Area

The Yellow River is among the earliest cradles of Chinese civilization and the origin of Chinese agricultural culture. The Yellow River Basin encompasses the geographical ecological region affected by the river system from its source to its estuary, traversing sequentially from the west to the east through the Qinghai-Tibet Plateau, Inner Mongolia Plateau, Loess Plateau, and Huang-Huai-Hai Plain. The mainstream stretches approximately 5687 km in length, covering a basin catchment area of $81.3 \times 10^4$ km$^2$ [26]. It exhibits a high-to-low terrain gradient from west to east, with the middle segment characterized by severe soil erosion due to loess geomorphology. Loess Plateau is recognized as the largest ecologically fragile area in China [26,55]. The lower reaches are characterized by low-lying topography, primarily consisting of the Yellow River alluvial plain. The YRB features a monsoon climate known as the "rain and heat period", marked by ample sunlight, high temperatures, and abundant precipitation. These conditions present superior climate resources, serving as a significant natural foundation for the development of agricultural civilization.

### 2.2. Ancient Channels of the Lower Yellow River and Its Influential Region

Historical records confirm that the lower reaches of the Yellow River have experienced over 1500 instances of flooding and more than 20 significant river diversions since the pre-Qin era [56,57]. The ancient river channel of the lower Yellow River demonstrates a tendency to shift back and forth between the Huai River and the Hai River (represented by purple lines in Figure 1). Moreover, the Huang-Huai-Hai Plain has been historically encompassed within the YRB. Variations in the lower course of the Yellow River, determined by its geographical path and estuary, can be categorized into northern, eastern, and southern flows.

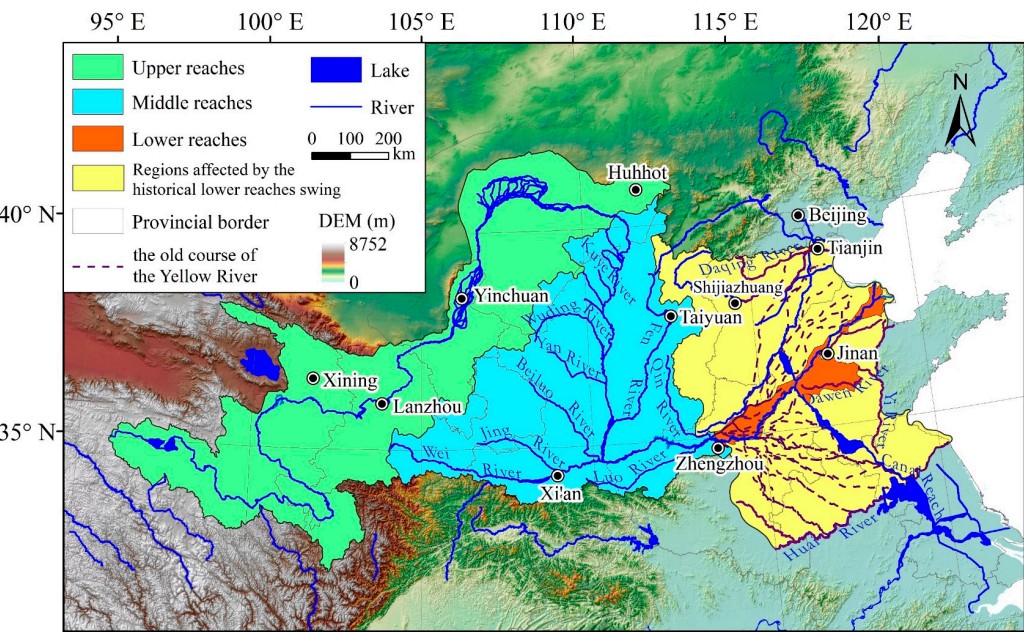

**Figure 1.** Geographic extent of the Yellow River Basin and schematic diagram of identified ancient river channels in the lower reaches. The modern-defined lower reach of the river is depicted in red, and the area influenced by channel swing in the lower course in history is highlighted in yellow.

In the past millennium, the northward flow occurred during the late Northern Song Dynasty (1048–1128 AD), merging with the Hai River system as it passed through the Hebei Plain and entered the Bohai Sea on the west coast. Subsequently, a southward flow occurred from the second year of the Jianyan period of the Southern Song Dynasty (1128 AD) to the fifth year of the Xianfeng period of the Qing Dynasty (1855 AD), as the

lower reach merged with the Huai River and discharged into the Yellow Sea. Since 1855 AD, the river has followed an eastward flow, passing through the ancient Ji River and Luo River basins before reaching the Bohai Sea, thus forming the current configuration of its lower reach. Therefore, the lower Yellow River region in this paper refers to the expansive area affected by channel swing in the lower course in history (highlighted in yellow in Figure 1), exceeding the modern-defined lower reach of the river (highlighted in red in Figure 1) greatly. Specifically, this study precisely delineated the historical lower Yellow River boundary as the vast area north of the Huai River, stretching to the Daqing River in the middle of the Hai River Basin and eastward to the Yishui and Mi River on the Shandong Peninsula. Thus, the scope of this study encompasses the contemporary Yellow River Basin and the region impacted by the oscillation of the downstream channel.

*2.3. Data Sources and Processing of Historical Land Cover Data*

After defining the historical extent of the YRB, we utilized ArcGIS 10.5 software to generate the corresponding shapefile data. Then, the shapefile data was employed as a mask to enable the batch extraction of land cover data in TIFF format using the Extract by Mask function within Arcpy. This study focused on investigating three prominent land cover categories: cropland, forestland, and grassland. These categories were chosen due to their extensive coverage, notable dynamic changes, and mutual transformations among land classes. Moreover, the availability of historical gridded data was taken into account. The records of settlement land coverage in historical documents are relatively limited compared to those of the three mentioned land types. Additionally, the complex driving mechanisms and diverse forms of land use change pose significant challenges in reconstructing historical spatial-temporal patterns of settlement land. Therefore, the construction land in history is not involved in this study.

2.3.1. Historical Cropland Cover Data

The cropland gridded data from 1000 AD to 2000 AD (Figure 2) was obtained from He et al. [16]. Through extensive analysis of historical records, the researchers constructed a provincial cropland area dataset employing methods including coefficient calibration, per capita cropland estimation, and proxy data conversion. They further assessed the land suitability for cultivation by incorporating altitude, slope, and agroclimatic potential productivity. This time-series dataset objectively reveals the spatial distribution of cropland from 1000 to 2000 AD, depending on the reliable historical evidence and validated reconstruction model.

2.3.2. Historical Forest Cover Data

We acquired the grid data of forest coverage in history (Figure 3) from Yang's research [34]. By analyzing the existing provincial forest area and population data over the past three centuries, an empirical Logistic regression model and an inverted sigmoidal log-linear relationship between the mentioned two variables were established. Then, the study estimated the provincial forest area from 1000 AD to 2000 AD by taking the provincial population as a proxy for anthropogenic deforestation activities. It was observed that areas with favorable natural conditions were cultivated first in China's land reclamation history. As a result, deforestation prioritizes lands with higher suitability for cultivation, gradually spreading to marginal lands with lower suitability. Based on this knowledge, a spatially explicit reconstruction model was developed to generate 10 km resolution maps of forest coverage in China from 1000 to 2000 AD.

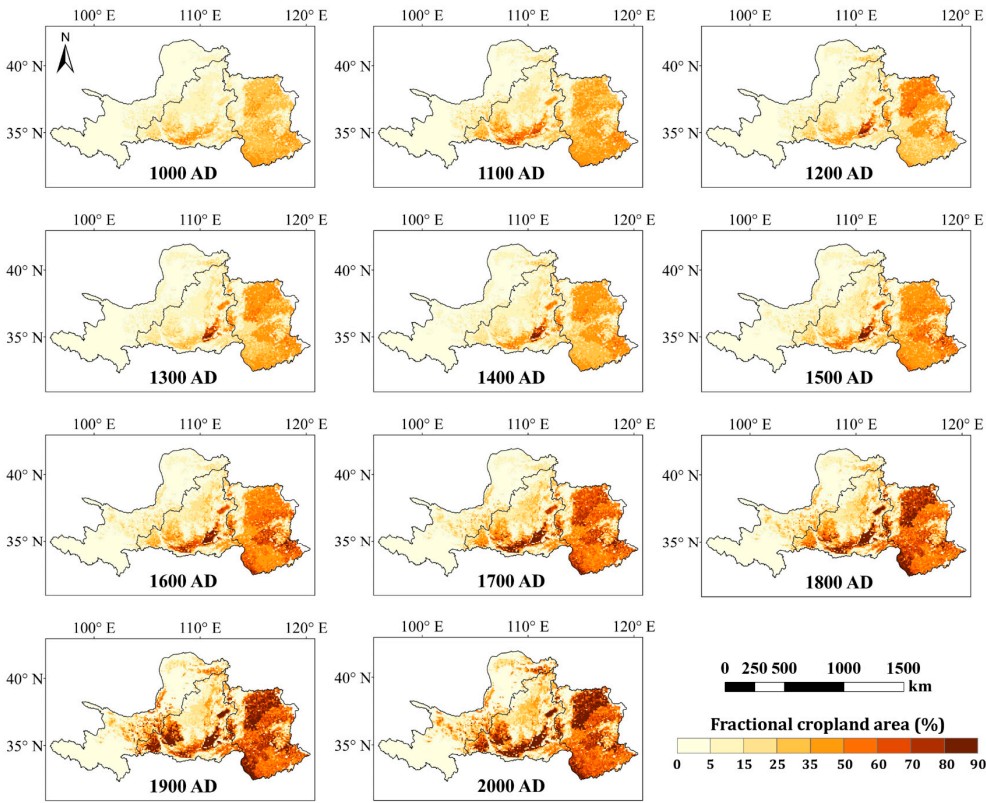

**Figure 2.** Cropland cover in the YRB from 1000 to 2000 AD.

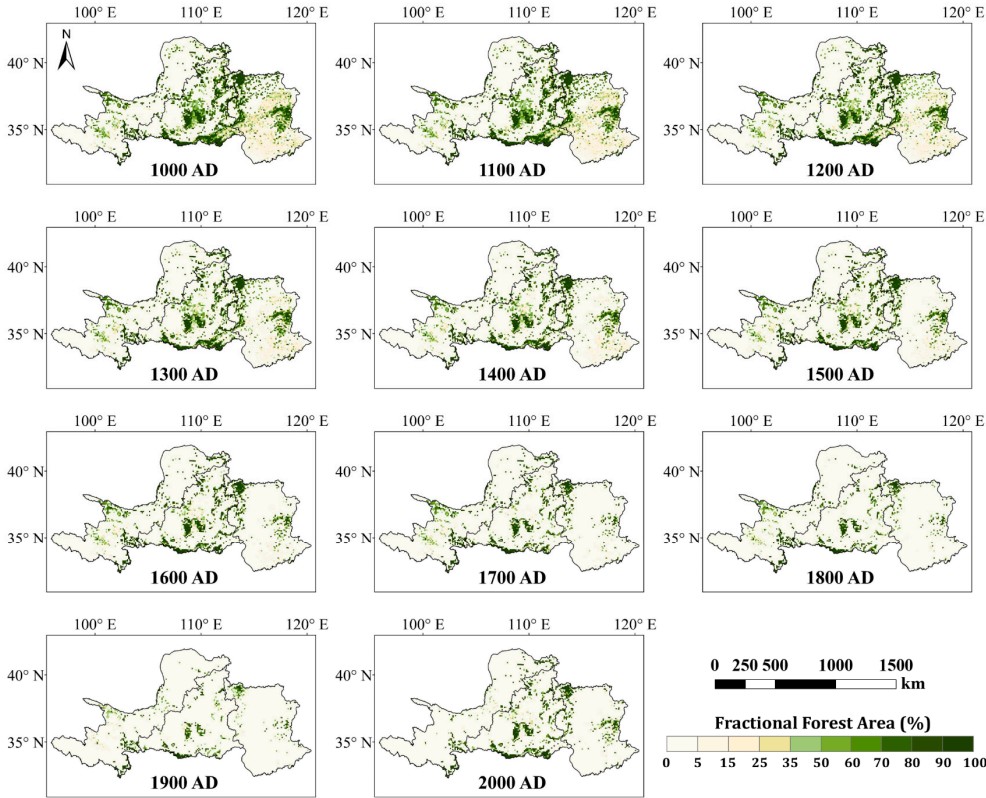

**Figure 3.** Forest cover in the YRB from 1000 to 2000 AD.

### 2.3.3. Historical Grassland Cover Data

We referenced the historical grassland cover data (Figure 4) from Yang's research [34]. In Western China, the researcher determined the historical grassland distribution by subtracting reconstructed historical cropland from the potential extent of natural grassland. In Eastern China, the habitat range of modern secondary grassland served as a limit of historical grassland distribution. By subtracting the forest and cropland cover from the overlapped area between the potential extent of natural forest and secondary grassland, Yang created the historical secondary grassland cover in a 10 km resolution and estimated the area subsequently.

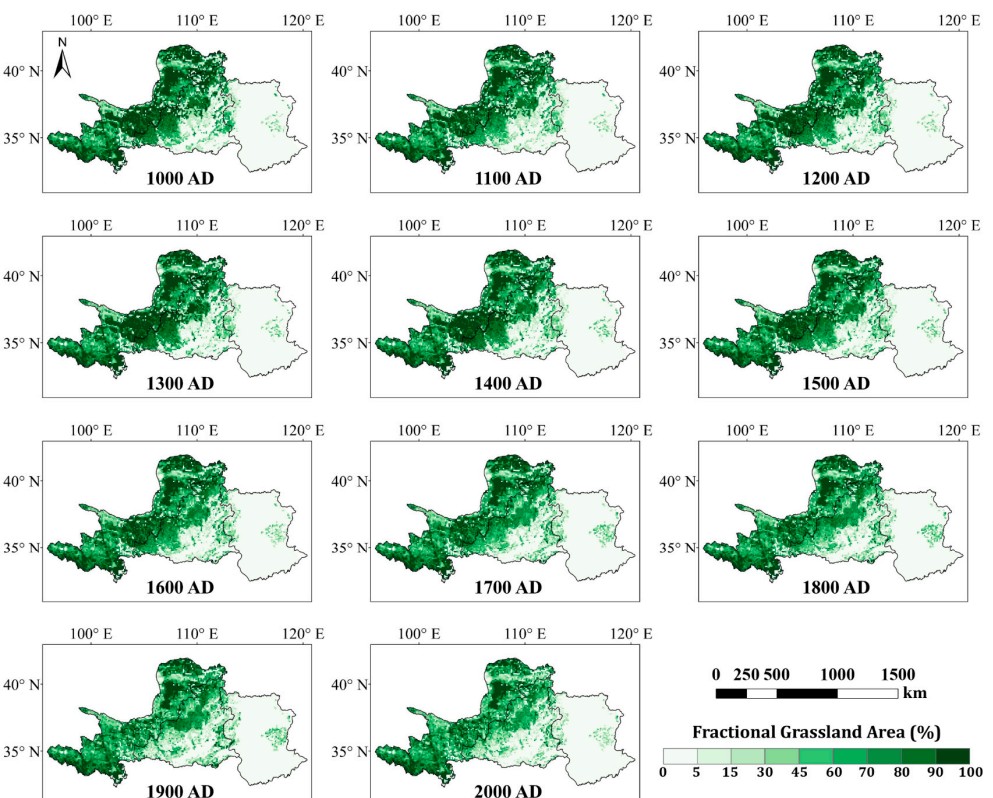

**Figure 4.** Grassland cover in the YRB from 1000 to 2000 AD.

## 3. Methods

### 3.1. Spatial Analysis of Kernel Density

Kernel Density illustrates the spatial concentration of cropland, forest, and grassland. In the context of land cover change analysis, Kernel Density analysis can help in understanding the spatial distribution patterns and clustering degree of different types of land cover. By examining the density of specific land use types or alterations in land use categories across a landscape, we can identify and visualize the hotspots of land cover distribution, the trends, and key areas of land cover change in different geographic areas over time. We first convert raster data to point data using the ArcGIS Conversion Tool, then calculate the density of point features around each output raster cell. In kernel density analysis, points within the search area are assigned varying weights based on their proximity to the search center. Points closer to the search center receive higher weights, and vice versa, lower weights. The density is determined by the following formula:

$$Density = \frac{1}{(radius)^2} \sum_{i=1}^{n} \left[ \frac{3}{\pi} \cdot pop_i \left( 1 - \left( \frac{dist_i}{radius} \right)^2 \right)^2 \right]$$

$$For \ dist_i < radius$$

(1)

in this equation, $i$ (1, 2, 3, ......, $n$) represents the input points; $pop_i$ is the value of point $i$; $dist_i$ denotes the distance between point $i$ and position $(x, y)$; and radius refers to the search radius. To determine the default search radius, we sequentially calculate the mean center of the input points, the distance from the mean center for all points, the median of these distances $(D_m)$, and the Standard Distance $(SD)$. The grid code (transformed from the raster value) is selected as the Population field to calculate the weighted items mentioned above. Subsequently, the following formula is employed to calculate the search radius:

$$\text{Search} Radius = 0.9 * \min(SD, \sqrt{\frac{1}{\ln(2)}} * D_m) * n^{-0.2} \tag{2}$$

where $n$ is the sum of the population field values.

Finally, the weighted distance is calculated as follows:

$$SD_w = \sqrt{\frac{\sum\limits_{i=1}^{n} w_i(x_i - \overline{X}_w)^2}{\sum\limits_{i=1}^{n} w_i} + \frac{\sum\limits_{i=1}^{n} w_i(y_i - \overline{Y}_w)^2}{\sum\limits_{i=1}^{n} w_i} + \frac{\sum\limits_{i=1}^{n} w_i(z_i - \overline{Z}_w)^2}{\sum\limits_{i=1}^{n} w_i}} \tag{3}$$

where $w_i$ is the weight of $i$; $\{\overline{X}_w, \overline{Y}_w, \overline{Z}_w\}$ signifies the weighted mean center.

### 3.2. Indicators for Cropland Development and Depletion of Forest and Grassland

The indicator used in this study assesses the degree of cropland development and the depletion of forest and grassland. The development degree quantitatively reflects the extent of actual new development occurring in a specific land use type within a certain time period, while the depletion degree evaluates the consumption or reduction within the same context. The formula is as follows:

$$D_{ab} = U_b - U_a \tag{4}$$

where $D_{ab}$ denotes the changed area of cropland, forest, and grassland in the YRB from time $a$ to time $b$; $U_a$ is the initial area of the corresponding land use type at time $a$, while $U_b$ represents the final area at time $b$.

### 3.3. Gravity Center Transfer Model

The geographic gravity center serves as an indicator of the spatiotemporal distribution characteristics of a specific geographical element. The direction and distance of the centroid's movement provide insights into the magnitude of changes and spatial variations of the geographical element over a defined period. In this study, the land use gravity center transfer model is employed to illustrate the migration of the gravity center of cropland, forest, and grassland in terms of distance, orientation, and trajectory. The equation is as follows [58]:

$$\overline{X} = \sum_{i=1}^{n} X_i S_i / \sum_{i=1}^{n} S_i \tag{5}$$

$$\overline{Y} = \sum_{i=1}^{n} Y_i S_i / \sum_{i=1}^{n} S_i \tag{6}$$

where $\overline{X}$ and $\overline{Y}$ indicate the latitude (lat) and longitude (lon) coordinates of the gravity center of one certain land use type; the variable $n$ denotes the number of spatial units; $X_i$, $Y_i$ represents the lat and lon coordinates of the $i$th spatial unit, respectively; $S_i$ stands for the area of this land use type in $i$th spatial unit.

The migration distance of the gravity center $(D_{s-k})$ between two years $(s, k)$ is calculated using the formula:

$$D_{s-k} = C \times [(Y_s - Y_k)^2 + (X_s - X_k)^2]^{1/2} \tag{7}$$

here, $(X_s, Y_s)$ and $(X_k, Y_k)$ represent the coordinates of the gravity center in the year of s, k, respectively; the constant C (111.111) serves as a conversion coefficient to translate the geographical coordinates into the plane distance (km).

## 4. Results and Analysis

### 4.1. Changes in Cropland, Forest, and Grassland Area in Yellow River Basin from 1000 to 2000 AD

Figure 5a illustrates the dynamic changes in different land cover types in the YRB over the past millennium. In the year 1000 AD, cropland covered approximately 11.65% (125,578.70 km$^2$) of the total YRB area, while forested areas comprised about 21.67% (235,264.58 km$^2$), and grasslands occupied approximately 41.69% (449,242.47 km$^2$). From 1000 to 2000 AD, the proportion of land under cultivation showed a noticeable overall growth, reaching 29.97% (324,659.38 km$^2$) by 2000 AD, indicating an 18.32% increase. Conversely, the forest area exhibited a decreasing trend during the same period, declining to 8.64% (93,738.71 km$^2$), a decrease of 13.04%. The area covered by grasslands remained relatively stable, accounting for 35.85% (388,344.86 km$^2$) in 2000 AD, representing a 5.83% reduction. Notably, the transformation of forests into farmland was most evident during this time period.

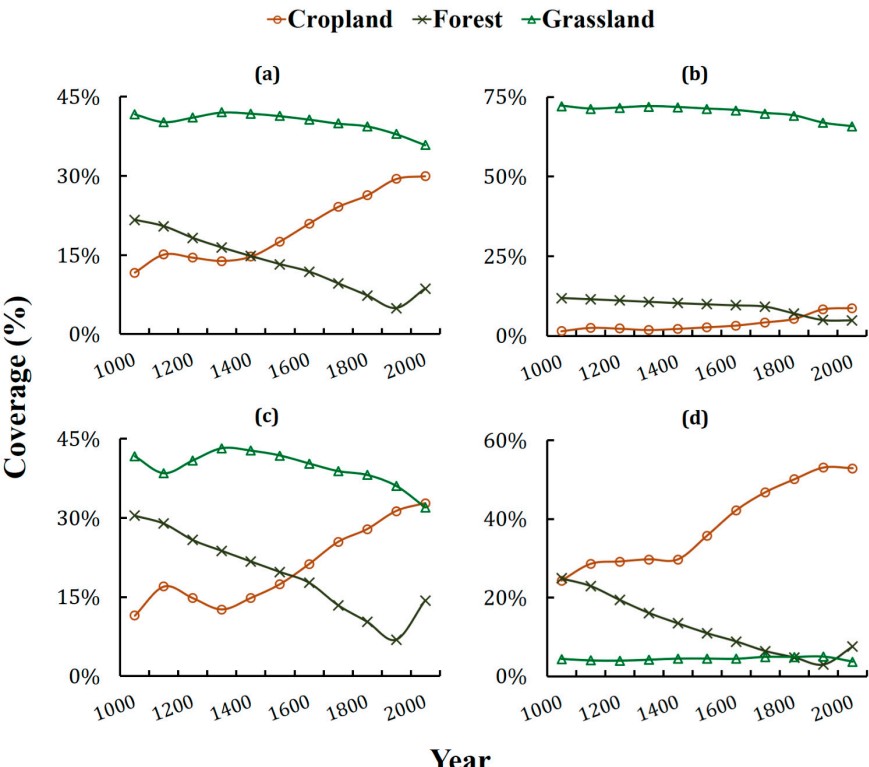

**Figure 5.** Changes in cropland, forest, and grassland coverage over the past millennium. Subfigures (**a**–**d**) depict the overall changes in the entire YRB and the specific changes in the upper, middle, and lower reaches, respectively.

Figure 5b,c depicts the variations of cropland, forest, and grassland coverage across the upper (b), middle (c), and lower (d) regions of the YRB from 1000 to 2000 AD. In the upstream areas, the overall cropland coverage shows an upward trend, increasing from 1.46% to 8.63%. From 1500 to 2000 AD, there was a marked acceleration in cropland expansion, with the annual increase averaging 685.85 km$^2$ between 1800 and 2000 AD, achieving a significant leap. In contrast, the forest area experienced a general decline, with coverage decreasing from 11.87% to 4.89%. The stages of change in forested areas parallel those observed in cropland. The grassland coverage trend remains relatively stable. Despite an overall downward trend, grassland continues to be the predominant land type in the upstream regions.

The characteristics of changes in the midstream show the closest resemblance to the entire YRB when compared to the other two regions, manifesting an augmented cropland area at the expense of diminishing forest and grassland areas. Initially, grassland dominates the landscape. Forested areas persisted in decreasing until pre-1900 but displayed some recovery thereafter. In contrast, cropland experienced a downturn between 1100 and 1300 AD, followed by a sustained upsurge in proportion. During 1500–1600 AD, cropland coverage surpassed forest for the first time, and by 2000 AD, it exceeded grassland once more, emerging as the predominant land cover in the middle reaches.

The proportion of cropland area in the downstream region has consistently been the largest, steadily increasing from 24.27% in the year 1000 AD to 52.85% in 2000 AD. The transformation of forested areas into cropland stands out as the most pivotal form of land cover change. Between 1400 and 1900 AD, there was a notable increase in cropland area, with an average annual increment of 161.59 km$^2$. Concurrently, the forested area underwent a drastic reduction, plummeting to its lowest coverage of 2.95% in 1900 AD before recovering to 7.55% by 2000 AD. Overall, the long-term historical accumulation in the mid-to-lower reaches of the YRB from the Northern Song Dynasty to the present evidenced a discernible trend of ecological deterioration and excessive depletion of forest resources. These trends have had significant negative impacts on livelihoods and socio-economic development.

Figure 6 illustrates the variations in the proportions of cropland, forest, and grassland areas across the upper, middle, and lower reaches of the YRB relative to the total cropland, forest, and grassland area over the past millennium. In the entire YRB, approximately 60% of cropland is concentrated in the lower reaches. This percentage has exhibited fluctuations ranging from 55.53% to 64.84% over the past millennium.

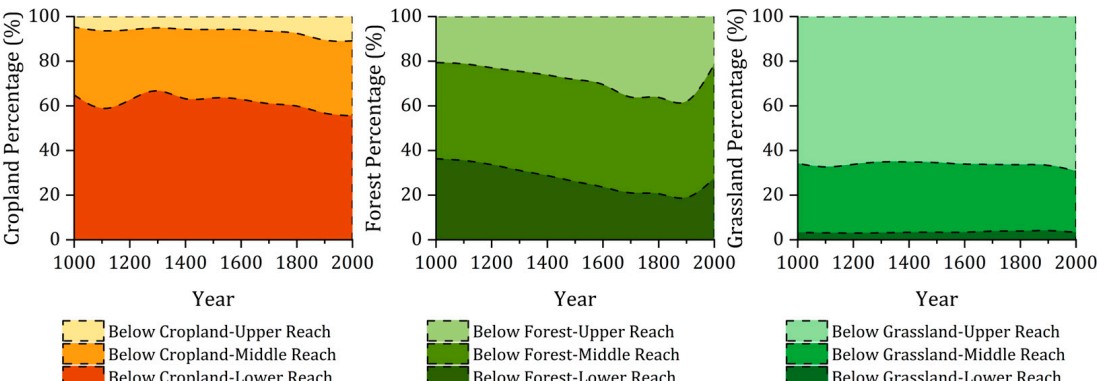

**Figure 6.** The changes of proportions of cropland (forest, grassland) area within the upper, middle, and lower reaches concerning the total cropland (forest, grassland) area in the YRB over the past millennium.

The distribution of forest land across the upper, middle, and lower reaches demonstrates relative uniformity. Specifically, in the upper reaches, the proportion of forest land escalated from 20.66% in 1000 AD to a peak of 38.03% in 1900 AD and subsequently receding to 21.35% in the ensuing century. At the same time, the trend in the downstream area is exactly the opposite. In the middle reaches, the forest land proportion oscillated between 43% and 51%. Combined with Figure 5, it becomes evident that all reaches experienced a declining trajectory in forest cover, with the analysis indicating that the lower reaches underwent more pronounced deforestation compared to the upper and middle regions.

About 70% of grassland is distributed in the upstream of the YRB. In comparison to cropland and forest land, the proportion of grassland within the three sub-regions remains relatively constant. Considering the declining trend of grassland areas in the upper and middle reaches, it is evident that more grassland has been converted into cultivated land, particularly in the midstream than in the upstream.

From 1000 AD to 2000 AD, the notable transformation of forests into cropland was particularly pronounced within the YRB. Nationally, the focus of land reclamation during this epoch predominantly gravitated towards the mid-to-lower reaches of the Yellow River and its counterpart, the Yangtze River. In the former region, the land reclamation rate surged from 15.35% to 36.54%, marking a growth of 1.38 times over a millennium. Meanwhile, the latter region experienced an increase from 14.89% to 27.32%, expanding by 0.83 times [16]. According to research conducted by Yang (2020) [34], by 1950 AD, the forest coverage in various regions of China—North China, Southeast China, Southwest China, Northeast China, and Northwest China—had dwindled to 14%, 36%, 32%, 48%, and 35%, respectively, compared to their forest coverage in the year 1000 AD. Consequently, the North China region downstream of the Yellow River emerges prominently in Chinese history as one of the earliest areas developed for agriculture, undergoing the earliest deforestation and experiencing the most severe environmental degradation. The land cover changes in the YRB stand as the most emblematic case of historical land cover alterations in China. Research on this subject holds significant importance for understanding the historical evolution of land cover changes in China.

### 4.2. Spatial Analysis of Land Cover Change over the Past Millennium across the Yellow River Basin

In this section, a comprehensive spatial analysis is conducted to scrutinize the alterations in land cover within the YRB spanning the last millennium. Data from five centennial years (1000, 1300, 1500, 1800, and 2000 AD) are selected to correspond with significant historical epochs, including the Song, Yuan, Ming, and Qing dynasties, alongside the contemporary era. Figure 7 is a timeline illustrating the evolution and succession of Chinese dynasties over a millennium, from 1000 to 2000 AD. Notably, in this section, we delve into the exploration of driving factors influencing changes in land cover. This analysis is informed by a consideration of the social development processes and characteristics unique to various historical periods.

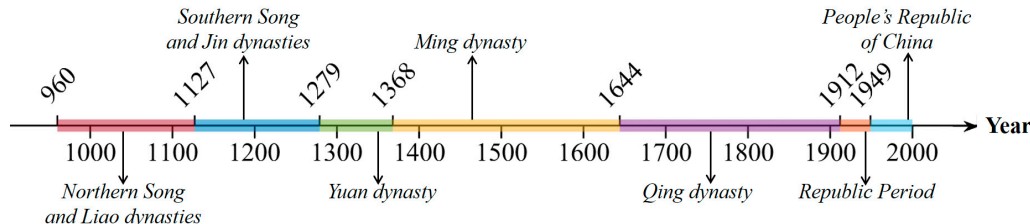

**Figure 7.** Dynastic Transitions of China: 1000–2000 AD.

#### 4.2.1. Spatiotemporal Characteristics of Cropland Cover Changes

Figure 8 illustrates the kernel density variations in the distribution of cropland within the YRB from 1000 AD to 2000 AD. Changes in cropland cover across different dynasties have been synthesized with historical records for an in-depth analysis of the driving factors, including economic, social, and environmental perspectives.

From the years 1000 to 1300 AD, spanning the Song, Liao, and Jin periods, the cropland coverage rate in the YRB increased from the initial 11.65% to 15.08% in 1100 AD and then gradually decreased to 13.86% by 1300 AD (Figure 4). During the Northern Song dynasty (960–1127 AD), the cultivation expanded northwestward, primarily through the practice of "Tuntian stationed" in the upper and middle reaches. Borderland soldiers actively reclaimed land near forts to secure food sources, leading to unprecedented agricultural development in the Loess Plateau areas. At that time, the river valley areas emerged as the primary regions for settlement farming, owing to their relatively flat topography, fertile soil, and convenient proximity to water sources that facilitated irrigation. Simultaneously, the downstream region of the Yellow River experienced frequent river channel breaches and floods. In response, local inhabitants developed a method called "yu-tian", involving the diversion of the river to accumulate silt and the reclamation of fields after sedimentation.

The organic material in silt proved to be highly beneficial for crop growth, signifying the successful transformation of adversity into advantage by the local population.

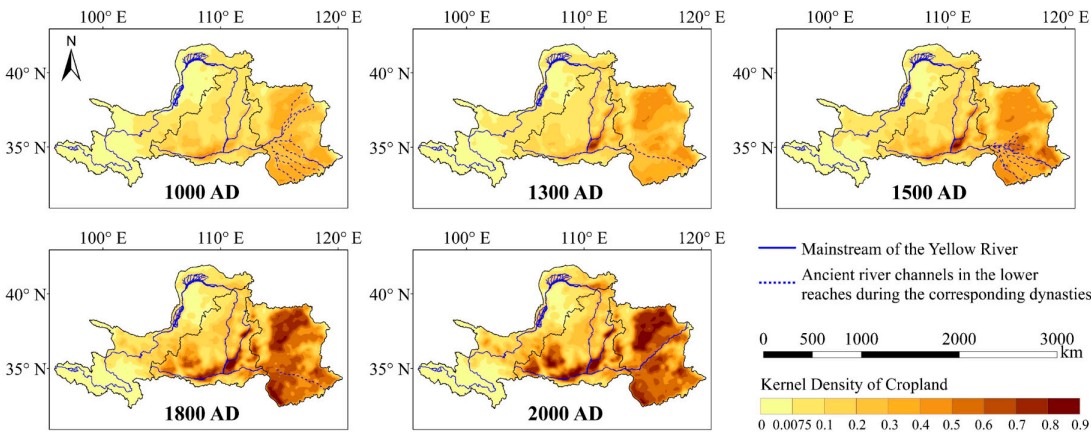

**Figure 8.** The kernel density variations of cropland within the YRB in 1000, 1300, 1500, 1800, and 2000 AD.

By 1200 AD, conflicts between the Northern and Southern Song dynasties resulted in land abandonment and a decline in cultivation rates in the mid-to-lower reaches, notably in the Wei River plain and the north of the Huai River (Figure 9). The subsequent Song-Yuan wars intensified this phenomenon. Extensive fallow land resulting from wars served as a conducive environmental foundation for settlement farming during the Yuan Dynasty.

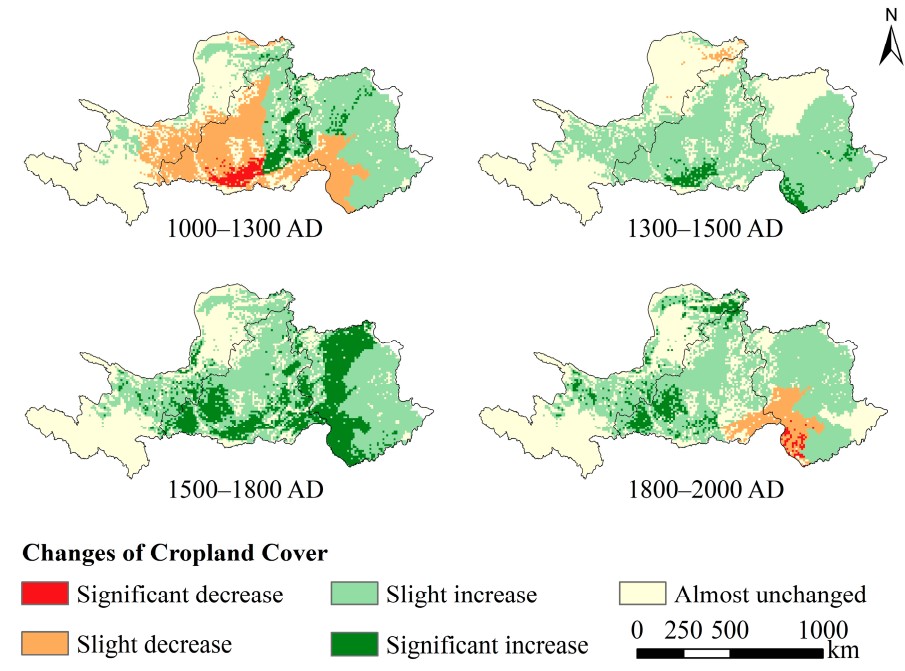

**Figure 9.** The evolution of the degree of cropland development across various historical periods.

Between 1300 and 1600 AD (spanning the Yuan and Ming periods), the cultivated area in the YRB witnessed a consistent expansion, experiencing an average annual growth of 255.58 km². The cropland coverage rate exceeded 20% for the first time, surging from 13.86% to 20.94%. During the Ming Dynasty, the military exigencies prompted a substantial focus on administration in the northwest region of the Hexi Corridor. This period was marked by robust endeavors in immigration and settlement, culminating in the emergence of "military-driven settlement farming nationwide, predominantly in the northwest." The northern segment of the Loess Plateau witnessed the largest military settlement farming

area, peaking at 54,000 hectares (approx. 5 million mu). By 1600 AD, a substantial portion of the sloping land in the Loess Plateau and the Hehuang Valley in the upper Yellow River had gradually been cultivated. The growth in population, increased societal demands, and advancing cultivation techniques collectively prompted the further expansion of new arable land.

From 1600 to 2000 AD, vast mountainous areas in the YRB were extensively developed, with cropland coverage rates increasing from 20.94% to 29.97%, marking an almost 10% growth. In the Qing Dynasty, there was a notable surge in land reclamation rates in the downstream regions of the YRB (Figure 9), approaching saturation by the 18th century. By 1900 AD, the level of cropland development in upstream areas, including the Liupan Mountain region in Gansu Province, the Ningxia Plain, and the Hehuang Valley, had almost reached the level observed in the downstream plains (Figure 8). Notably, cultivation expanded from the plains to mountainous areas during this period.

This shift primarily stems from two factors: (1) the introduction and promotion of American crops. China experienced rapid population growth during the Qing Dynasty, yet the expansion of cultivated land failed to keep pace with the burgeoning population, exerting significant pressure on the food supply. The introduction and promotion of American crops like corn and sweet potatoes played a pivotal role in addressing the escalating demand for food. They often exhibit characteristics such as high yield, resistance to infertility, and cold tolerance, facilitating the cultivation of previously challenging terrains like mountains and low-lying areas. By the end of the Qing Dynasty, these crops accounted for over 20% of China's grain production. (2) Incentives arising from the Qing government's policy on cultivating mountainous land. In 1740 AD, an imperial edict was promulgated, elevating the previously overlooked and deemed uncultivable "mountainous regions" to priority cultivation areas. This edict included tax exemptions, rendering it highly appealing to farmers and precipitating the influx of impoverished populations into these mountainous areas. Concurrently, advancements in water management initiatives facilitated more farmland for crop cultivation and agricultural production. The YRB solidified its position as a pivotal agricultural hub in China.

4.2.2. Spatiotemporal Characteristics of Forest Cover Change

Over the past millennium, the forest coverage in the YRB gradually diminished, with a brief recovery observed only by the end of the study period. Figure 10 illustrates the fluctuations in the kernel density of forest coverage, while Figure 11 presents variations in the degree of forest depletion during different historical periods within the basin.

From 1000 to 1400 AD, the forest coverage in the YRB experienced a decline from 21.67% to 16.41%. Historical documents indicate that during the Song and Yuan dynasties, wood consumption for people's livelihoods, such as cultivation, construction, fuel, and other purposes, ultimately contributed to a recognizable clearance of forests.

As depicted in the kernel density map (Figure 10), forest coverage was primarily concentrated in the mountainous areas of the midstream in the year 1000 AD, during the Northern Song dynasty. Meanwhile, natural forest vegetation in the lower reaches of the North China Plain has disappeared due to human land reclamation. Apart from deforestation for cultivation, the Northern Song dynasty extensively utilized wood for construction materials and fuel. The *Extended Continuation to Zizhi Tongjian* reports that, around 1043 AD, Dongjing (the capital city of the Northern Song dynasty) annually consumed 300,000 logs for construction purposes. The *Wenxian Tongkao* documents that in i1065 AD, "a total of 17.13 million units of firewood and one million units of charcoal were transported from Jingxi, Shaanxi, and Hedong to Dongjing." Such intense logging often exceeds the natural recovery capacity of the forests, leading to a significant retreat of forest distribution in most areas of Shaanxi Province, eastern Gansu Province, and west of Liupan Mountain. Prolonged conflicts between the Song, Jin, and Western Xia regimes occurred in proximity to the Liupan Mountains. The extensive military operations and agricultural practices in

this area inflicted severe harm upon the expansive forests, emerging as one of the primary catalysts behind the alterations in Liupan Mountain's forest landscape.

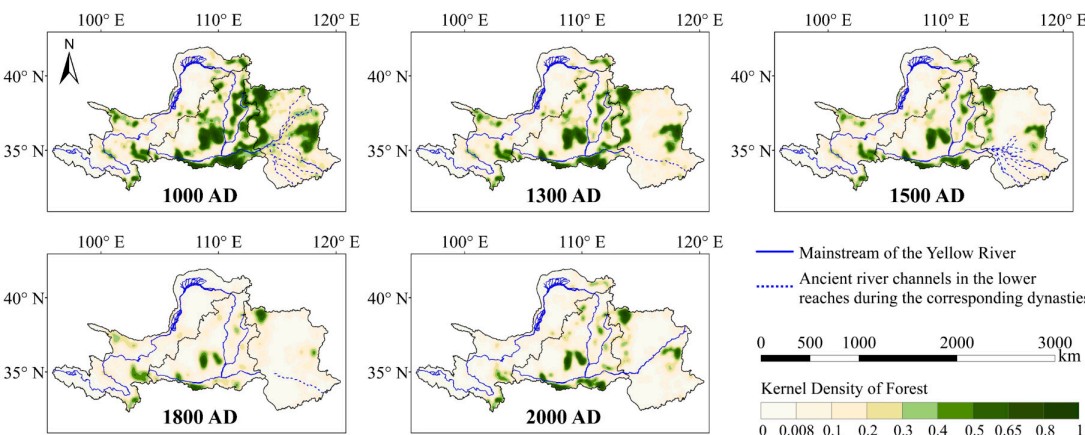

**Figure 10.** The kernel density variations of forest within the YRB in 1000, 1300, 1500, 1800, and 2000 AD.

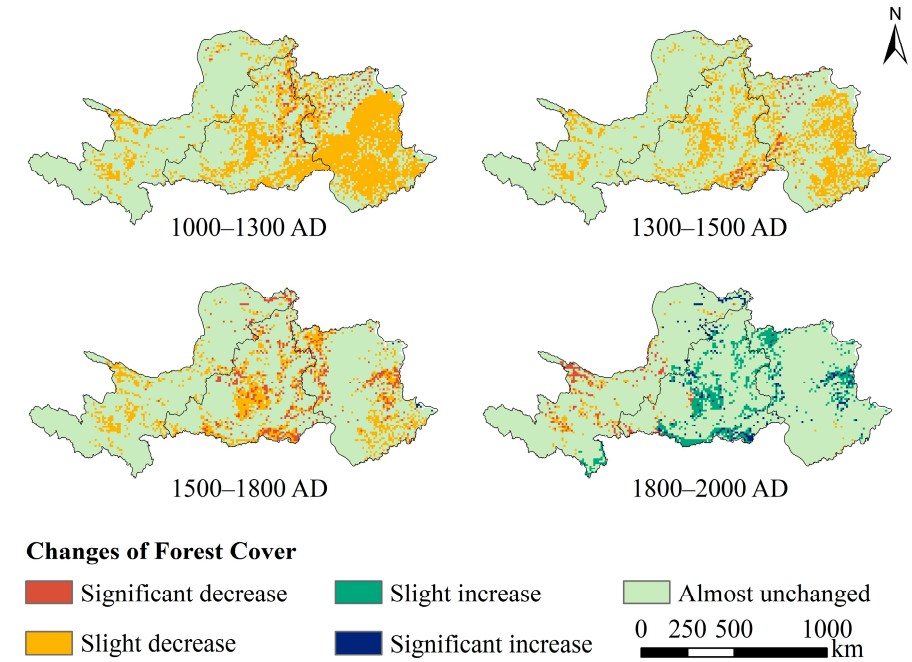

**Figure 11.** The evolution of the degree of forest depletion across various historical periods.

Extensive deforestation in the YRB intensified notably during the Ming and Qing dynasties, particularly since the latter half of the Ming Dynasty. From 1400 to 1900 AD, forest coverage in the region plummeted from 16.41% to 4.93%. The reasons for the precipitous decline in forests coincide with the sharp increase in cultivated land. Both dynasties ardently advocated for land reclamation towards mountainous regions to mitigate the food supply pressure caused by the burgeoning population growth. Besides, the introduction and cultivation of maize played a pivotal role. Despite the infertile terrain of mountainous regions, maize's resilience to drought and cold enabled its growth in high-altitude and rocky landscapes. These multifaceted influences precipitated mass migrations to mountainous areas. These migrants, predominantly motivated by agricultural pursuits, exhibited little regard for ecological conservation, resulting in extensive deforestation and environmental deterioration. The destruction of indigenous vegetation in mountainous regions often leads to significant soil erosion, ultimately resulting in nutrient depletion.

As a consequence, cultivation becomes unsustainable, prompting farmers to eventually abandon their land and relocate elsewhere. As illustrated in Figure 3, the forest coverage in the YRB exhibited significant spatial sparsity in 1900 AD. Prior to the establishment of the People's Republic of China, the forest vegetation in the lower reaches had been nearly eradicated.

Since 1900 AD, years of warfare, followed by intense industrialization and urbanization, have led to a decline in forest cover in the YRB to below 5%. Following the establishment of the People's Republic of China, governmental initiatives promoting afforestation and forest protection successfully restrained widespread logging in this region, thereby catalyzing a gradual recovery of vegetation. By 2000 AD, there had been an obvious improvement in forest coverage, increasing from 4.93% in 1900 AD to 8.64%.

### 4.2.3. Spatiotemporal Characteristics of Grassland Cover Change

Figure 12 illustrates the kernel density changes of grassland in the YRB over the past millennium, while Figure 13 shows the variations in grassland depletion levels across distinct historical periods. In contrast to significant spatial changes in cropland and forest cover, the trend in grassland alteration seems comparatively stable.

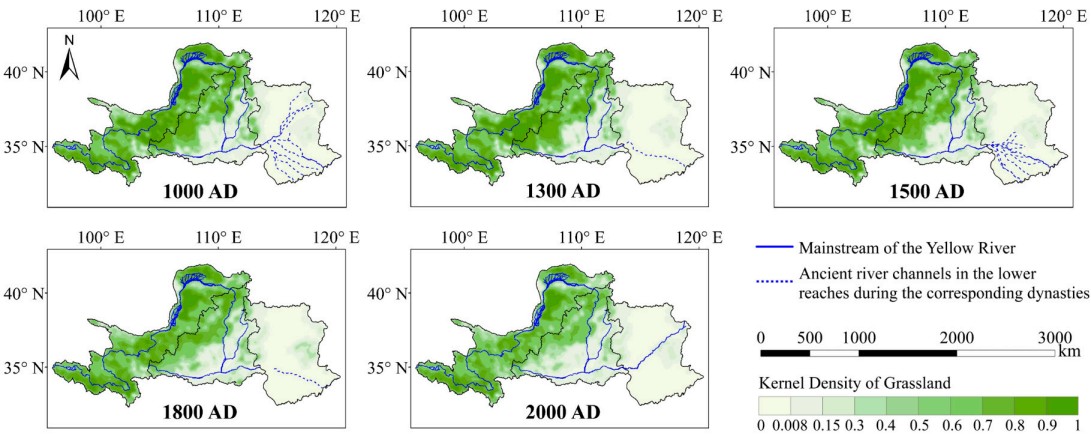

**Figure 12.** The kernel density variations of grassland within the YRB in 1000, 1300, 1500, 1800, and 2000 AD.

From 1000 to 1200 AD, the overall grassland coverage initially decreased from 41.69% to 40.22%, subsequently rebounded to 42.04% by 1300 AD. The upstream of the Yellow River region represents one of the primary distribution areas for grassland in China. Prior to 1300 AD, during the Song, Liao, and Jin periods, this region was governed by Western Xia and Tubo regimes. The economic activities of the Tubo tribes were predominantly centered on animal husbandry, with agriculture playing a secondary role in the economic structure. Consequently, the extensive "fertile fields", as perceived by the inland Han people, remained fallow during that period. The pastoral areas of Western Xia boasted favorable conditions with abundant water and grass, particularly in regions like Ganzhou, Yanzhi Mountain, and the plains of Yinchuan. These areas are characterized by deep water and thick soil, providing robust support for lush vegetation. At that time, agriculture thrived only in the Hetao Plain, Ningxia Plain, and the Hehuang Valley. Wars and abrupt population decline often provide the environment with a respite, facilitating recovery without human interference, thus somewhat decelerating the process of ecological transformation. When combined with Figure 9, it becomes evident that the abandoned cropland resulting from the Song-Yuan wars in the upper and middle reaches swiftly reverts to grassland without human intervention.

From 1300 to 1800 AD, the grassland coverage in the YRB dropped from 42.04% to 39.42%. During the Yuan Dynasty, the Mongols, as nomadic people, preserved specific areas to establish pastures for grazing. This practice of converting certain lands into pastures

effectively precluded their reuse for cultivation, thereby ensuring the diversity in land cover. In the early Ming Dynasty, numerous cultivated lands ravaged by war were abandoned and left fallow due to neglect in cultivation. This was especially true in the Guanzhong Plain in the midstream region and the North China Plain in the downstream region. Thorns and other secondary vegetation swiftly spread across these regions.

Before extensive land reclamation activities during the Qing Dynasty, grasslands predominated in the Hehuang Valley. Following the Qing Dynasty, the extent of natural vegetation in this region gradually decreased, giving rise to a predominantly agricultural-pastoral zone. Figure 13 shows a marginal recovery of grassland in the middle reaches during the mid-Qing Dynasty. At that time, the introduction of crops appropriate for cultivation in mountainous regions, such as corn and potatoes, coupled with the influx of a large population into the mountains due to livelihood pressures, resulted in severe deforestation and land clearance. However, due to the constraints of the natural environment, the land fertility in mountainous areas is insufficient to sustain long-term cultivation. A significant portion of arable land was abandoned after losing fertility. Some of these regions that were previously covered by forests underwent a conversion to grasslands through secondary succession. As a result, there was an observable rise in grassland coverage in mountainous areas of the middle and lower reaches from 1500 to 1800 AD (Figure 13).

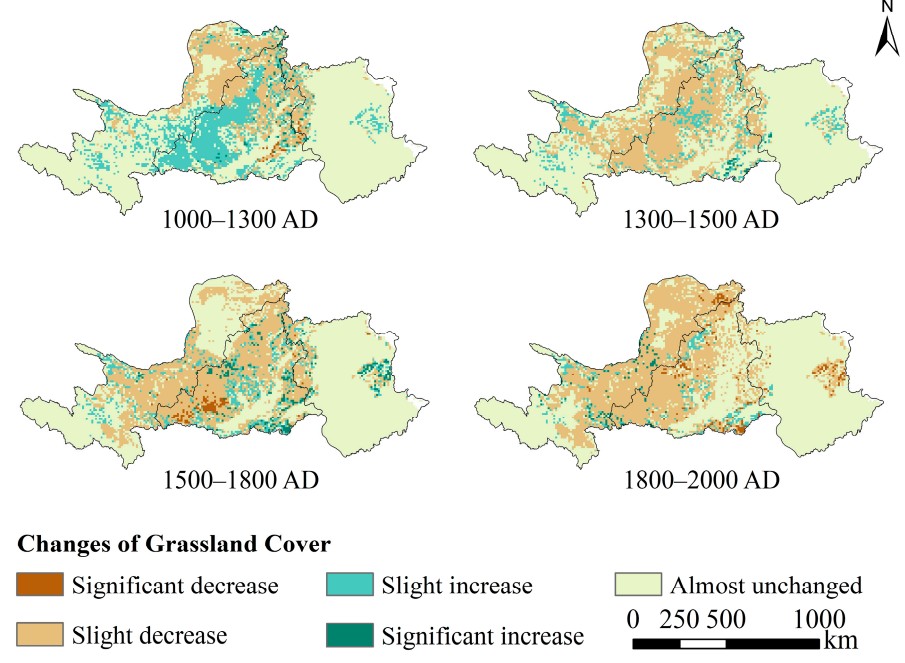

**Figure 13.** The evolution of the degree of grassland depletion across various historical periods.

Since 1800 AD, the grassland area in the basin has undergone a rapid decline, decreasing to 35.85% by 2000 AD, with an average annual reduction of 192.88 km². Potential factors contributing to this decline encompass intensive cultivation, overgrazing, and the recent accelerated urbanization, all of which promote the degradation of grasslands and alterations in land cover. Furthermore, forest vegetation has progressively regenerated as a result of government initiatives such as afforestation and forest protection, which has resulted in a sustained reduction in the extent of secondary grasslands affected by human activities.

### 4.3. The Gravity Center Migration of Cropland, Forest, and Grassland

In this investigation, we computed the spatial gravity centers of grassland, forest, and cropland within the YRB over the last millennium (Figure 14). The results indicate that they are distributed in the upstream, midstream, and downstream regions, respectively. All gravity centers of the three land cover types exhibit a westward migration trend during

the studied period. Notably, no significant inter-regional migration was observed for the gravity centers.

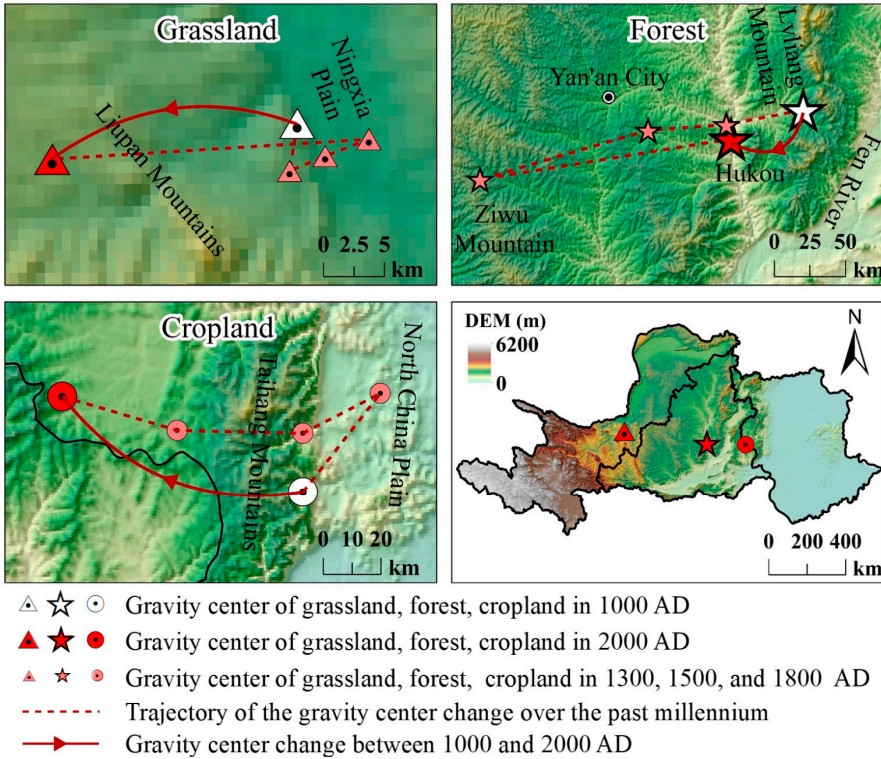

**Figure 14.** The gravity center migration of cropland, forest, and grassland over the past millennium. In the legend, the icon from left to right represents grassland, forest, and grassland, respectively. The solid line depicts the net migration distance of the gravity center between 1000 and 2000 AD, while the dashed line illustrates the general migration trajectory of the gravity center over the last millennium.

The gravity center of grassland is located at the junction of the Ningxia Plain and the Liupan Mountains area in the upstream region. Between 1000 and 2000 AD, the center underwent a westward shift of approximately 20.25 km. During this period, the migration distance was shorter from 1000 to 1800 AD, whereas the westward migration distance from 1800 to 2000 AD extended to 25.95 km. In contrast to forest and farmland, this alteration is less dramatic. The spatial centroid of the forest is situated in the Hukou region, at the junction of Lvliang Mountain and Fen River. Over the past millennium, it has steadily shifted southwest, covering a net distance of 54.74 km. Between 1000 and 1800 AD, the forest gravity center consistently migrated westward, relocating near the Ziwu Mountains in 1800 AD, approximately 232.65 km from the initial center in 1000 AD. From 1800 to 2000 AD, the forest center reversed its migration, returning to the Hukou area. The trajectory of the forest gravity center exhibits the broadest east-west span. The spatial gravity center of cropland is concentrated at the junction of the North China Plain and the Taihang Mountains. During the period from 1000 to 1300 AD, farmland migrated 43.57 km in the northeast direction. Thereafter, it consistently shifted westward and, by 2000 AD, moved northwest (compared to 1000 AD) to the boundary between the upper and lower reaches. This migration extended further into the Taihang Mountains, with a net migration distance of 89.78 km.

The migration direction of cropland and forest indicates that, during the Ming and Qing dynasties, cropland in the lower reaches of the YRB had reached saturation, gradually expanding towards the middle reaches. Land reclamation in the middle reaches primarily occurred at the cost of the consumption of forest in mountainous areas, exacerbating soil erosion in the Loess Plateau. The "Interim Outline for Soil and Water Conservation of the People's Republic of China," issued in 1957, proposed the prohibition of cultivating

on sloping land and afforestation and grass planting on retired cropland. Subsequent measures aimed at restoring the vegetation of the Loess Plateau have achieved significant success. These measures could partially explain why the migration of the forest gravity center shifted back in 2000 AD.

## 5. Discussion

### 5.1. The Main Achievements of This Study

The research has clearly delineated the geographical extent affected by channel swing in the lower course of the Yellow River in history. Specifically, the delineated area extends from the north of the Huai River to the Daqing River in the middle of the Hai River Basin and eastward to the Yishui and Mi Rivers on the Shandong Peninsula. Based on this, we employed various methods, including kernel density analysis, the development and depletion degree of land cover, and the gravity center transfer model. These approaches were utilized to conduct a systematic analysis, incorporating timing, positioning, and quantification of the spatiotemporal characteristics of land cover changes within the YRB over the past millennium for the first time. Then, we select five specific time points (1000, 1300, 1500, 1800, and 2000 AD) to investigate the variations in land cover across distinct historical periods (Song, Yuan, Ming, Qing, and modern era). Finally, we analyzed the general mutual conversion processes among cropland, forest, and grassland and examined the driving forces behind changes in land cover across different historical periods.

### 5.2. The Relationship between Population Growth, Cropland Expansion, and Environmental Changes

In the preceding Section 4, we have comprehensively expounded upon the driving forces that have shaped alterations in land cover within the YRB across various dynasties. The fluctuations in cropland coverage within the YRB stem from the integrated impact of diverse factors. Population growth and advancements in agricultural technology both play pivotal roles in the expansion of cropland. Furthermore, the evolution of water conservancy projects additionally enhances the security of water resources for farmland, thereby facilitating land reclamation.

In the early Northern Song Dynasty (1000 AD), both population and agricultural development in the YRB were in their initial stages but relatively limited. The disturbance of the original vegetation from anthropogenic activities was primarily concentrated in areas such as river valleys and plains. With the gradual increase in population, agricultural development in the YRB advanced rapidly, and the distribution of cultivated land continually expanded outward from the southeast of the Loess Plateau and the southwest of the North China Plain [59,60]. At that time, the cultivated land in the YRB accounted for 30.2% of the national cultivated land, with an average reclamation rate of 21%. In the plain areas of the mid-to-lower reaches, such as the Huang-Huai-Hai and Guanzhong regions, the reclamation rates mostly exceeded 40. During the Yuan Dynasty, a significant portion of land in the middle reaches underwent transformation into pastoral zones, contributing to a certain degree of recovery in natural vegetation. Following the Ming Dynasty, there was a swift resurgence and advancement in agricultural cultivation within the middle and lower reaches. By this time, the majority of hilly terrains had also been reclaimed.

The Ming and Qing periods witnessed the zenith of population growth in China. Confronting the challenges of "*insufficient food and land*," the government enacted a series of measures to incentivize the population to reclaim land in less cultivable "*mountainous regions*". The implementation of tax exemptions made these regions particularly attractive to farmers, leading to a notable influx of impoverished populations into these previously overlooked areas. Although these measures partially alleviated the pressure on food, the irrational development of mountainous areas induced various environmental problems. Extensive cultivation resulted in the depletion of forests and grasslands, accompanied by soil erosion, water and soil loss, floods, and water resource issues. After environmental degradation, soil fertility declined, leading to a decrease in agricultural productivity.

Consequently, individuals migrated to regions with preserved environments and higher productivity. Thus, population migration is not only a cause but also a consequence of environmental changes, establishing a mutual causal relationship between the two.

### 5.3. The Recovery of Forest and Grassland

Over the past millennium, forest coverage in the YRB has consistently decreased, primarily attributed to factors including agricultural development, population growth, economic progress, and resource utilization demands. Since the 1950s, the Chinese government has intensified its commitment to environmental protection, implementing measures (e.g., "*grain for green*" and "*hillclosing afforestation*") to stop the creeping encroachment on forests and grasslands. Consequently, forest coverage has commenced increasing in certain mountainous regions. The restoration of forested areas mainly depends on afforestation projects initiated after the establishment of the People's Republic of China. By 2000 AD, more than 30 million hectares of forests have been planted in northern China.

The current secondary grasslands arise from the degradation of vegetation on deforested sites under specific habitat conditions following repeated instances of forest destruction. They underwent a developmental process, transitioning from woodland to cultivated land and eventually transforming into secondary grassland. Two reasons can explain the increase in grassland coverage: (1) Population decline resulting from warfare often affords the environment a temporary respite. As the recovery of woodlands typically lags behind that of grasslands, farmlands abandoned due to war-induced disruptions are more likely to undergo secondary succession, ultimately transforming into grasslands in the absence of human interference. For example, after the Song-Yuan War, there was a transient resurgence in grassland coverage in the southwestern region of the middle reaches around 1300 AD, compared with the conditions observed in 1000 AD (Figure 13). (2) The reclamation of mountainous areas during the Ming and Qing periods usually ceased once the soil fertility became depleted. Therefore, abandoned cropland in these mountainous regions underwent secondary succession, evolving into herbaceous shrub communities. This transition is evident in the form of grassland recovery, as observed between 1500 and 1800 AD (Figure 13).

### 5.4. The Disparity between the Expanded Cropland Area and Diminished Forest and Grassland Area

Between 1000 and 2000 AD, the disparity between the expansion of cropland and the reduction of forest and grassland covers an area of 3342.80 km$^2$. This value may be even higher during specific historical periods. It indicates that besides being converted into cropland, forests and grasslands have also transformed into other land use types, serving diverse purposes such as extensive construction materials and fuel. In recent times, particularly since urbanization and industrialization, the conversion of forests and grassland has extended to broader areas earmarked for urban and rural residential construction, exacerbating the shrinkage of forests and grasslands. Two noteworthy points emerge. Firstly, the net increase in arable land may arise not only from the conversion of forest and grassland but also from the cultivation of previously uncultivated land. Secondly, a significant portion of urban and rural construction land is transformed from arable land, and regions originally adorned with natural vegetation may undergo the sequence of "forest/grassland-arable land-construction land." Both factors can lead to the net difference (3342.80 km$^2$) being substantially smaller than the actual value. Hence, a more comprehensive analysis is imperative for understanding the specific quantitative relationships between cropland, forest land, and grassland conversion.

Furthermore, considering the reconstructed data itself, the choice of a gridded spatial allocation model for cropland data reconstruction may introduce errors due to the model's inherent "tiling" defect. Similarly, simulating potential vegetation distribution during forest and grassland data reconstruction may differ in accuracy from natural vegetation reconstructed through archaeological methods (pollen, tree rings). These factors contribute

to uncertainty in the reconstructed data, and these uncertainties and errors propagate into the data analysis. This necessitates the incorporation of additional historical land cover data and rigorous, detailed historical records. Meanwhile, strengthening the integration with archaeological data, such as collecting and organizing various ancient vegetation records like pollen and tree rings, would enhance data comparison and interpretation.

## 6. Conclusions

A comprehensive analysis of land cover changes in the YRB over the past millennium has provided insights into the spatiotemporal dynamics of cropland, forest, and grassland. Through this period, cropland coverage in the YRB increased from an initial 11.65% to 29.97%, whereas forest and grassland coverage decreased from 63.36% to 44.49%. The spatial distribution of cultivated land consistently expanded outward from the southeast of the Loess Plateau and the southwest of the North China Plain. The concentration of cropland in downstream areas, the gradual decline in forest coverage, and the restoration of secondary grassland highlight the intricate interaction between human activities and environmental changes. The impact of land cover changes shapes the ecological and societal landscape of the YRB. The prioritized cultivation of fertile lands and the subsequent expansion into less suitable areas indicate historical and socioeconomic factors that have molded the region's development.

These findings underscore the intimate link between human activities and environmental changes. Extensive cropland development has led to reductions in forest and grassland areas, triggering a cascade of environmental issues, including soil erosion, water and soil loss, and heightened vulnerability to floods. Environmental degradation, in turn, affects population migration, with communities typically seeking refuge in areas with better environmental quality and higher agricultural productivity. To address the issue of vegetation degradation, modern government initiatives such as afforestation programs and sustainable land management practices have been implemented. These measures have facilitated the gradual restoration of forest and grassland cover, showcasing the potential for positive change through strategic conservation efforts.

In conclusion, this study offers a nuanced understanding of the historical trajectory of land cover changes in the YRB. By combining quantitative data with qualitative analysis, we delve into the profound impact of human activities on the ecosystem of the region, advocating for additional measures to achieve sustainable land use and environmental protection. Meanwhile, these analyses lay the foundation for quantifying the environmental impacts of land cover changes for future research.

**Author Contributions:** Conceptualization, Y.W., F.Y. and F.H.; data, F.Y. and F.H.; methodology, Y.W. and F.Y.; formal analysis, Y.W.; writing—original draft preparation, Y.W.; writing—review and editing, Y.W., F.Y. and F.H.; funding acquisition, F.Y. and F.H. All authors have read and agreed to the published version of the manuscript.

**Funding:** This research was funded by the National Natural Science Foundation of China (Grant No. 42201263) and the National Key Research and Development Program of China (Grant No. 2017YFA0603304).

**Institutional Review Board Statement:** Not applicable.

**Informed Consent Statement:** Not applicable.

**Data Availability Statement:** Data is contained within the article.

**Conflicts of Interest:** The authors declare no conflicts of interest.

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
