# Peer review of "Spatiotemporal Characteristics of Land Cover Change in the Yellow River Basin over the Past Millennium"

_land, doi:10.3390/land13020260_

Round 1

Reviewer 1 Report

Comments and Suggestions for Authors

Manuscript ID: land-2867586

Title: Spatiotemporal Characteristics of Land Cover Change in the Yellow River Basin Over the Past Millennium

This manuscript provides sufficient information in the background section as well as in other sub-headings. Moreover, the research design is appropriate and can be replicable for further investigation. Thus, this manuscript is suitable after performing some major revisions.

Abstract: The study time table or period should be defined .

Introduction: line 54, number six indicates what?..... Land Cover 6k.

Information provided between line 76 to 81 requires citations. For instance, what is the rate of population growth? And the rate of forest and grassland destruction should be supported with numerical information.

Data sources

This study selected three land cover categories, namely, cropland, forestland, and grassland, for investigation. Because cropland, forestland, and grassland represent the most extensive coverage with notable dynamic changes and mutual transformations among land classes. What about settlement land cover classes?

Results are ok, but the authors not able to discuss their results with some scientific literature. Thus, I recommend the authors to include discussion section as a separate sub-heading or they can combine results and discussion.

Conclusion is weak. The authors repeat the results. Therefore, this section requires major revisions. The authors can conclude based on their results rather than reporting results in percentages.

Author Response

Please see the attachment 1.

Reviewer 2 Report

Comments and Suggestions for Authors

This study focuses on the temporal and spatial characteristics of land cover change in the Yellow River Basin (YRB) over the past Millennium, and discusses the ecological and environmental impacts behind its evolution. Overall, this research is excellent in many ways, but there are still some potential flaws that could affect the depth and breadth:

Introduction: The introduction of the manuscript effectively highlights the importance of understanding historical land use and land cover change (LUCC) in the context of the Yellow River Basin (YRB). The introduction could be improved by providing a more detailed background on the current state of research in the YRB, including any recent studies that have influenced the field. Additionally, a clearer articulation of the research gaps that this study aims to fill would strengthen the introduction.

Literature Review: The literature review section is comprehensive and well-referenced, covering a wide range of studies on LUCC, historical climate change, and the carbon cycle. The authors have done a commendable job in synthesizing the existing knowledge and identifying the need for spatially explicit representations of dynamic LUCC in the YRB. However, the review could benefit from a more critical analysis of the methodologies used in previous studies, particularly those related to gridded reconstruction of land cover. This would help to contextualize the authors' approach and its potential advantages or limitations.

Methods: The methodology section is detailed and transparent, outlining the use of kernel density analysis, development and depletion degree indicators, and the gravity center transfer model. The authors have clearly defined the study area and the data sources, which is crucial for replicability. However, the manuscript would benefit from a more thorough explanation of the rationale behind the chosen methods, especially the kernel density method, and how it was adapted to the specific context of the YRB. Additionally, a discussion on the potential limitations of the methods, such as the accuracy of historical data and the assumptions made in the reconstruction process, would provide a more balanced view.

Results: The results section presents a clear and systematic analysis of the spatiotemporal changes in land cover in the YRB. The use of visual aids, such as the migration trajectories of land cover gravity centers, effectively communicates the findings. The authors have successfully quantified the degree of cropland development and the depletion of forests and grasslands. To enhance the impact of the results, the authors could consider comparing their findings with other regions or time periods to provide a broader context for the observed changes.

Discussion: The discussion section provides a thoughtful analysis of the driving forces behind the land cover changes, linking them to population growth, technological advancements, and policy shifts. The authors have also considered the implications of their findings for ecological conservation and sustainable development in the YRB. To strengthen the discussion, the authors could explore the potential feedback loops between land cover changes and climate change, as well as the socio-economic impacts of these changes. Additionally, suggestions for future research, such as the integration of more detailed historical records or the use of advanced modeling techniques, would be valuable.

Overall, the manuscript presents a significant contribution to the understanding of historical LUCC in the YRB. The authors have employed robust methods and provided a comprehensive analysis of the data. However, the review suggests that the introduction could be more specific about the research gaps, the methods section could benefit from a more critical evaluation of the chosen techniques, and the discussion could delve deeper into the broader implications of the findings and potential future research directions.

Author Response

Please see the attachment 2.

Round 2

Reviewer 1 Report

Comments and Suggestions for Authors

The authors addressed all previous comments and concerns.